# Meta-Analysis of Implementation Intentions Interventions in Promoting Physical Activity among University Students

**Sanying Peng** [1,2,*], **Ahmad Tajuddin Othman** [2], **Ahmad Zamri Khairani** [2], **Zhuang Zhou** [2], **Xiaogang Zhou** [2], **Fang Yuan** [3] and **Jinghong Liang** [4]

1 Department of Physical Education, Hohai University, Nanjing 210024, China
2 School of Educational Studies, Universiti Sains Malaysia, Penang 11800, Malaysia; judane@usm.my (A.T.O.); ahmadzamri@usm.my (A.Z.K.); zhouzhuang1994@outlook.com (Z.Z.); zhouxg@students.usm.my (X.Z.)
3 College of International Languages and Cultures, Hohai University, Nanjing 210024, China; yuanf@hhu.edu.cn
4 Department of Maternal and Child Health, School of Public Health, Sun Yat-Sen University, Guangzhou 510080, China; liangjh78@mail2.sysu.edu.cn
* Correspondence: pengsy@hhu.edu.cn

**Abstract:** Implementation intentions (II) have demonstrated their efficacy in promoting physical activity (PA) among various populations, including adults, the general population, and those with chronic conditions. However, there is a dearth of meta-analyses examining this efficacy among university students. This meta-analysis aims to determine the effectiveness of II interventions in improving PA among university students. Six online databases (PubMed, Embase, Web of Science, Cochrane Central, PsycINFO, and PsycArticle) were comprehensively searched. Recursive searching and grey literature collection strategies were employed to identify relevant studies. The meta-analysis employed a random-effects model to combine effect sizes for different PA outcomes. The Cochrane Risk of Bias tool was used to assess the quality of the included studies, and sensitivity analysis was performed to assess the robustness of the results. Twelve studies involving 1916 participants met the inclusion criteria. The findings indicated that II interventions significantly increased PA among university students compared to control groups (standardized mean difference [SMD] = 0.31, 95% confidence interval [CI]: 0.12, 0.50, $p < 0.001$). Subgroup analyses revealed larger effect sizes in the following groups: publication year after 2013, inactive university students, the reinforcement group, the action planning group, and the intervention period of fewer than six weeks. The above findings offer compelling evidence that II interventions can effectively promote PA among university students. The integration of IIs with e-health platforms and the implementation of individualized and multifaceted intervention models exhibit promising sustainability for promoting PA on campus.

**Keywords:** action planning; coping planning; implementation intentions; meta-analysis; physical activity; university students



## 1. Introduction

Physical activity (PA) is widely recognized as a crucial element in preventing chronic diseases and enhancing physical and psychological well-being [1]. Despite these well-established health benefits of PA, a significant proportion of individuals aged 10–24 years fail to meet the PA recommendations of WTO [2]. According to the survey, more than 81% of adolescents worldwide are considered physically inactive [3]. More alarmingly, participation in PA tends to decline with age [4], with a particularly sharp decrease during late adolescence and early adulthood, including among university students [5]. As such, promoting PA has become a top priority for university health agencies, and there is an urgent need for effective interventions to enhance the PA levels of university students.

Although many interventions have been employed to promote PA among university students, the synthesized results of previous systematic reviews and meta-analyses are

promising but inconsistent. For example, after combining the results of multiple types of PA interventions among university students, Plotnikoff et al. [6] found significant intervention effects and small effect sizes from the pooled results of 18 trials only for moderate PA, while no significant intervention effects were found for total PA. Favieri et al. [7] conducted a meta-analysis of 18 experiments that employed a range of interventions, including educational messages, brochures, motivational incentive courses, social media campaigns, and wearable tracking devices to promote PA. While the results of these interventions did not reach statistical significance, they did demonstrate a medium effect size. Based on information and communication technology (ICT), E-health has emerged as a promising measure for PA interventions. The behavior change techniques (BCTs) inherently embedded in e-health, such as goal setting, monitoring, and feedback mechanisms, are direct predictors of behavior change [8,9]. In this area, McIntosh et al. [10] conducted a systematic review, revealing that e-health has the potential to effectively improve PA in young adults. This finding is further supported by a recent meta-analysis conducted by Peng et al. [11], which demonstrates a marginally medium effect, particularly among university students. Additionally, a review advocated that PA interventions among university students should target their behavioral determinants and take an individualized approach with a model of session intervention [12]. Based on the above analysis, it is evident that BCT elements display a strong potential as effective and promising strategies for promoting PA among university students.

Implementation intentions (IIs), one of the crucial BCTs, is an explicit form of planning for a specific behavioral response to a target intention by developing the situational content that triggers the target behavior [13]. The conceptual framework of IIs is an if-then model that emphasizes that behaviors are triggered based on certain conditions, i.e., cues, but are inherently manifested in the form of action planning (AP) and coping planning (CP) [14]. AP pertains to an explicit mapping of when, where, and how to take action on a goal, while CP refers to the responses to possible obstacles that may arise in goal achievement [15]. Previous studies have revealed that IIs are determinants of behavior change [16,17] and that there is a definite mediating relationship between intentions and behavior [18,19], and some studies have verified that IIs can even have a moderating effect on the relationship between intentions and behavior as well [20,21]. The theory of planned behavior considers intentions as the most proximal predictor of behavior [22]. However, previous studies have identified a gap between intentions and behavior for certain complex behaviors [23,24], such as engaging in regular PA [25]. To fill the gap between intention and behavior, IIs have been proposed as a potential bridge, providing a rationale for their applications in facilitating behavior change [26].

Whilst previous meta-analyses have provided compelling evidence for the efficacy of IIs in promoting behavioral change [24,27–29], the translation of this approach to the context of PA has yielded disparate outcomes. For instance, a meta-analysis by Belanger et al. [30] revealed a significant small-to-medium effect size of IIs in promoting PA among adults. Similar findings have been reported recently in a general population [31]. Nevertheless, Carraro and Gaudreau [32] conducted a meta-analysis, which uniquely combined all relevant studies without restrictions on participant characteristics as both simultaneous and experimental elements, and found a smaller effect size in the pooled experimental data. These studies focused on populations with a wide age span, and such a combination method may contribute to high heterogeneity. A recent meta-analysis examining the effects of planning intervention on PA in a chronic disease population yielded positive results, but the pooled effect size was small [33]. Although the mixed findings mentioned above, IIs have been demonstrated to be a cost-efficient, feasible, and sustainable strategy for promoting PA across various populations [30,32,33]. This approach may be particularly suitable for university students who are at a crucial stage for developing healthy lifestyle habits [34]. However, there is a dearth of meta-analyses evaluating the effects of II intervention for increasing PA in university students, and no reports of subgroup analyses in this population have emerged from other relevant studies.

Therefore, this study has twofold purposes: first, to synthesize the effect sizes of implementation intentions on promoting PA among university students; second, to investigate the differences in intervention efficacy across moderators. The primary research question guiding this study is as follows: What is the effect of implementation intentions on promoting PA among university students, and to what extent does the intervention efficacy differ across various moderators?

## 2. Methods

This review was guided by the Cochrane Handbook [35] and PRISMA reports [36]. A preliminary search was conducted to determine the feasibility of this study based on three key inclusion elements, "implementation intentions", "physical activity", and "university students", and on this basis, registration was performed on the PROSPERO platform (CRD42023424579).

### 2.1. Search Strategy

A study examining the effects of implementation intentions on PA among university students was considered, without language restrictions and incorporating randomized controlled trials (RCTs). An exhaustive and comprehensive search was conducted in several databases, including PubMed, Embase, Cochrane Library, Web of Science, PsycINFO, and PsycArticle. The search process was performed using a combination of Medical Subject Headings (MeSH) and free terms with Boolean logical operators. The leading search terms included participants (university students, undergraduate, higher education students), intervention methods (implementation intentions, action planning, coping planning, planning), outcomes (physical activity, exercise), and study design (RCTs). In addition, a recursive manual search was performed by tracking references of similar systematic reviews or meta-analyses and reviewing studies presented at professional conferences to ensure comprehensive coverage and avoid missing crucial studies. Furthermore, unpublished grey literature was searched through specialized thesis databases to prevent the omission of essential studies. The supplementary material provides detailed information on the search strategies and search processes for all databases. All records collected above were imported into Endnote software for further compiling and management.

### 2.2. Eligibility Criteria

The primary inclusion criteria were as follows: (1) Enrolled university students were included as subjects in this study without other conditions. (2) Trials in which the intervention group received the implementation intentions interventions to promote any form of PA were included, whether step counting, moderate PA, vigorous PA, or total PA. (3) The outcomes of PA could be any form of measurement, including but not limited to indicators such as step count, energy expenditure, exercise time, number, and frequency. (4) The study design was an RCT, including pilot and cluster RCTs.

The exclusion criteria were: (1) Participants included university students with disabilities or mentally incompetent college students. (2) Specific measures and elements for the implementation intentions interventions were not specified in the study. (3) Studies with multiple interventions and a separate II intervention group could not be extracted. (4) Studies without control groups.

### 2.3. Data Extraction

After importing the retrieved data into Endnote 20 software (Thomson ISI Research Soft, Philadelphia, PA, USA), the duplicated records were first removed through an automated process, and the titles and abstracts of the remaining records were independently reviewed by the two authors (PSY and YF) of the study to determine the literature for full-text reading and inconsistencies and uncertainties were resolved by consulting a third reviewer (ATO). The following information was extracted from the included literature: (1) characteristics of the studies, including authors, publication year, sample size and

distribution, and region in which the trials were conducted; (2) information about the participants, including age, the proportion of females, and baseline PA level; (3) interventions details, including the specific model, intervention content, reinforcement, and duration; (4) information on outcomes, including the measurement instruments of PA and different outcomes of PA.

*2.4. Quality Assessment*

The risk of bias (ROB) for all included studies was assessed and classified using the 2nd version of Cochrane Risk of Bias tool [35], which consists of seven components: random sequence generation, allocation concealment, blinding of participants and personnel, blinding of outcome assessments, incomplete reporting of outcome data, selective reporting, and other biases. Review Manager (Version 5.4; The Cochrane Collaboration, The Nordic Cochrane Centre, Copenhagen, Denmark) was used for ROB evaluations. Each of the seven domains was evaluated independently, and the risk of bias was rated as high, low, or unclear for each of the included studies. In addition, funnel plots [37] and Egger tests [38] were employed to assess the presence of publication bias. The two authors (PSY and YF) of this review independently completed the quality assessment of each study according to the criteria guidelines. Disagreements and uncertainties in the assessment were adjudicated by consulting the other author.

*2.5. Statistical Analysis*

In light of the fact that the outcomes of included studies were continuous variables, and the units were inconsistent, standardized mean differences (SMDs) with 95% confidence intervals (CIs) were employed as the estimated effect sizes of meta-analysis [35]. Means and standard deviations of the comparison groups were extracted to calculate SMD. When other transformable values (e.g., standard errors) were reported in the study, they were transformed by statistical methods. The meta-analysis used a random effects model based on the inverse variance (DerSimonian–Laird method). $p$-values of Cochran's Q-test and $I^2$ were used to determine whether there was heterogeneity among the included studies, with $p < 0.1$ considered as significant heterogeneity, and values of $I^2$ statistics at 25%, 50%, and 75% considered as low, medium, and high heterogeneity of the cut-off values [39]. In addition, to explore the crucial influences of heterogeneity and to investigate whether the effect of II interventions varies across different contexts, subgroup analyses were performed in the following groups: Publication year (Publication year $\geq$ 2013 vs. Publication years < 2013); Participants (General vs. Inactive); Intervention strategy (AP vs. AP + CP); Reinforcement (Yes vs. No); and Duration ($\geq$6 weeks vs. <6 weeks). Sensitivity analyses based on stepwise elimination were used to assess the robustness of the pooled effect sizes. Data analysis was performed independently by the two authors, and disagreements and uncertainties were resolved through discussion and retrieval of the original studies. All calculations and analyses of the data above were conducted in STATA16 (Stata Corp, College Station, TX, USA).

## 3. Results

A comprehensive search was conducted across multiple databases, identifying a total of 3253 records for further evaluation. Following removing duplicates and incomplete trial records, 2675 articles were screened by title and abstract, with an additional 32 records identified through recursive search. From this process, 139 studies were selected for full-text reading screening, with an additional five Ph.D. theses retrieved through grey literature searching, yielding 144 studies for full-text review. Upon conducting a thorough full-text reading, 12 studies [40–51] were deemed suitable for quantitative meta-analysis after excluding non-RCT literature, studies that did not report reliable data, and studies that did not have outcomes aligned with the study topic. The entire process of literature screening is illustrated in Figure 1.

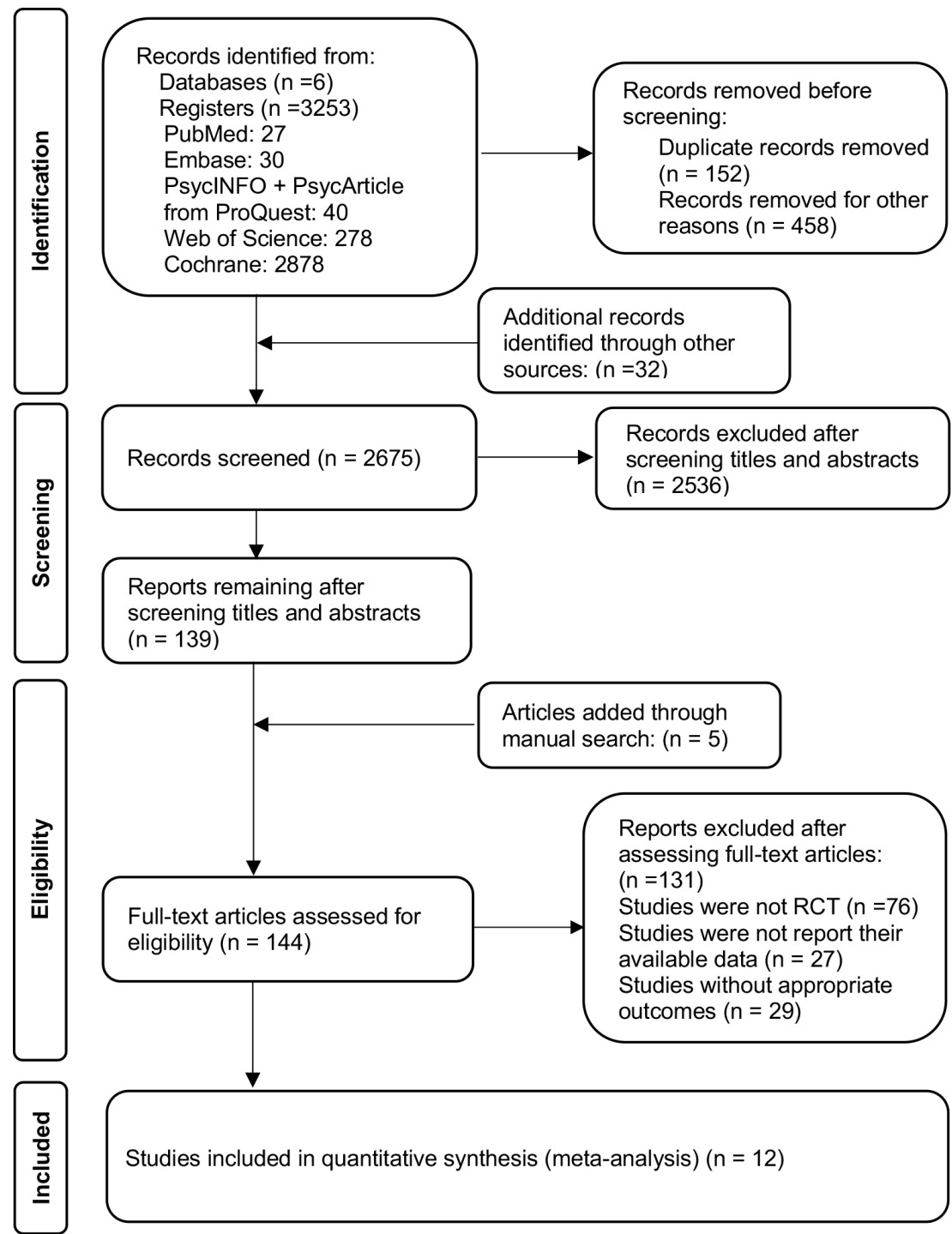

**Figure 1.** PRISMA flow chart of study selection.

### 3.1. Studies' Characteristics

Essential study characteristics comprised the authors, year of publication, sample size and distribution, and the region where the experiment was conducted. Three studies [49–51] were conducted within the last decade, and the other nine studies [40–48] were conducted ten years ago. All studies were conducted in developed countries, with half of the studies [40,41,45–48] being undertaken in the UK. There were four studies [42,44,50,51] conducted in North America, three [42,50,51] in the United States and one [44] in Canada. Two studies [43,49] were conducted in Asia, one each in Singapore [43] and Korea [49].

Information about the participants was represented by age, the proportion of females, and baseline PA characteristics of the participants. The mean age of the university students tested ranged from 18.2 to 23.76, the majority of the sample [40–47,49,50] (10/12) had a female proportion higher than 50%, four studies [43,45,50,51] targeted inactive university students who did not meet the PA recommendations, and the other studies did not have baseline requirements for the PA level of participants.

Details of the intervention implementation are as follows: first, most studies [40–46,49,50] used a single intervention model of AP, while only three studies [47,48,51] used a mixed intervention model of AP and CP. Then, all but two of the interventions were delivered in face-to-face sessions, except for two [47,51] that were delivered online. Finally, the follow-up period for the last measurement in the included studies spanned from 2 weeks to 11 months, during which five studies [44,45,49–51] applied reinforcement strategies in the form of mobile phone or email reminders and repeats of interventions session through face-to-face sessions.

Outcome information included the measurement instrument and the format of the results. The outcome measurement instruments were all self-reported questionnaires or items, except for one study [50] that used a pedometer when reporting baseline PA levels. The PA outcomes were presented in various forms, including the number of moderate-to-vigorous PA (MVPA) for a given time, the number of workouts, the frequency of movements, the number of hours of exercise per week, and energy expenditure.

The studies' characteristics are presented in Table 1.

### 3.2. Quality of Included Studies

All included studies were assessed as low risk on both selection and reporting bias. Three [40,41,45] were rated as high risk on performance bias, and two studies [47,48] were ranked as high risk on attribution bias. Most studies [40–50] were identified as an unclear risk for detection bias and other biases. Based on the risk evaluation criteria, one study [42] was identified as an unclear risk, and all other studies were identified as low risk. Quality summary for each item and the overall risk of bias for individual studies are shown in Figures 2 and 3.

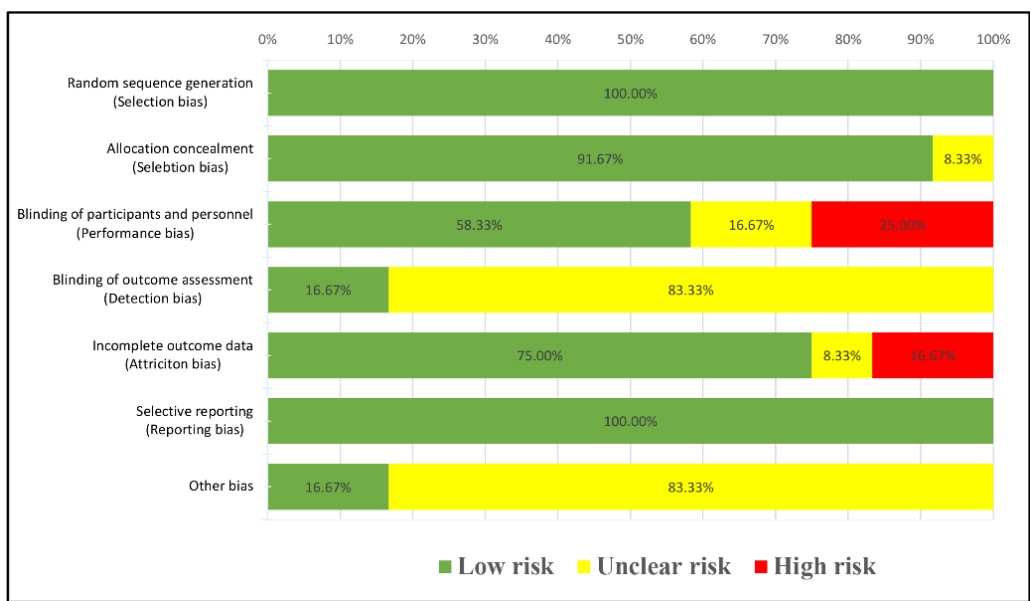

**Figure 2.** Risk of bias graph (each item presented as percentages).

**Table 1.** Characteristics of included studies.

| Publication | Sample Size | | Country | Sample Characteristics | | | | Intervention Characteristics | | | | Outcomes | |
|---|---|---|---|---|---|---|---|---|---|---|---|---|---|
| | IG | CG | | Age | | Female (%) | Participant | Intervention Content | Delivery Mode | Duration | Reinforcement | Instruments | Indicators |
| | | | | IG | CG | | | | | | | | |
| Milne et al., 2002 [40] | 79 | 93 | UK | 20.04 ± 2.23 | | 73 | General | AP | Session | 2 W | No | Items (not validated) | No. times MVPA > 20 min/ week |
| Prestwich et al., 2009 [45] | 29 | 34 | UK | 23.76 ± 4.64 | | 58.06 | Inactive | AP | Session | 4 W | Reminder by Mobile phone (not mentioned times) | Item (not validated) | No. times of MVPA > 30 min/ week |
| Zhang and Cooke, 2012 [48] | 22 | 21 | UK | 20.56 ± 1.62 | | 48.81 | General | AP + CP | Session | 4 W | No | Items (not validated) | No. times MVPA > 20 min/ week |
| Murray et al., 2009 [44] | 29 | 23 | Canada | 30.5 ± 9.8 | | 100 | General | AP | Session | 11 W | Three times repetition | Checklist at the gym (not validated) | No. sessions/week |
| Prestwich et al., 2003 [41] | 18 | 18 | UK | 21.31 ± 4.39 | | 51.2 | General | AP | Session | 4 W | No | Items not validated | No. sessions/week |
| Skår et al., 2011 [47] | 335 | 315 | UK | 22.8 ± 6.7 | | 63.4 | General | AP + CP | Online | 6 W | No | Items (validated) | Scores expressing frequency |
| Chatzisarantis et al., 2008 [43] | 92 | 35 | Singapore | 20.71 ± 6.95 | | 72.44 | Inactive | AP | Session | 5 W | No | LTEQ | Scores expressing frequency |
| Kim et al., 2019 [49] | 51 | 52 | Korea | 22.59 ± 1.77 | | 56.3 | General | AP | Session | 5 W | Repetition every week | Items (not validated) | Scores expressing frequency |
| Conner et al., 2010 [46] | 180 | 176 | UK | 20.7 ± 2.99 | | 69 | General | AP | Session | 2 W | No | Items (validated) | Frequency of exercise |

**Table 1.** *Cont.*

| Publication | Sample Size | | Country | Sample Characteristics | | | | Intervention Characteristics | | | | | Outcomes | |
| | IG | CG | | Age | | Female (%) | Participant | Intervention Content | Delivery Mode | Duration | Reinforcement | Instruments | Indicators |
| | | | | IG | CG | | | | | | | | |
| Waters, 2007 [42] | 54 | 60 | USA | 18.2 ± 2.0 | | 100 | General | AP | Session | 6 W | No | Items (not validated) | Mins of exercise/week |
| Bogg and Vo, 2022 [50] | 73 | 74 | USA | 20.56 ± 2.04 | | 60 | Inactive | AP | Session | 2 M | Three times by email, reminding | Pedometer + Items (not validated) | METs Exercise |
| Sur, 2022 [51] | 24 | 29 | USA | 19 | | N/R | Inactive | AP + CP | Online | 2 W | Repetition every day | IPAQ | Mins of MVPA/week |

Notes: AP: action planning; CG: controlled group; CP: coping planning; IG: intervention group; IPAQ: International Physical Activity Questionnaire; LTEQ: Leisure Time-Exercise Questionnaire; LTPA: leisure-time physical activity; M: month; METs: metabolic equivalents; MVPA: moderate to vigorous physical activity; W: week.

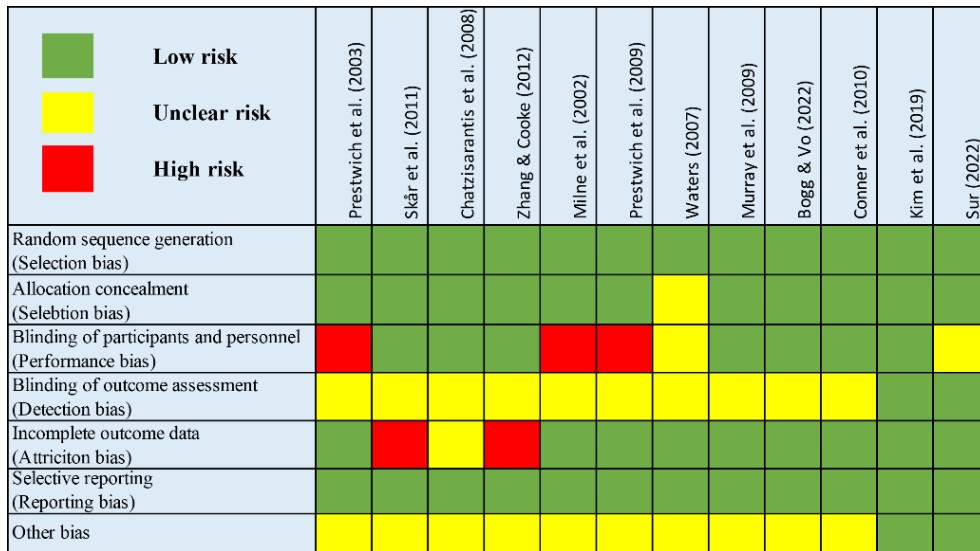

**Figure 3.** Risk of bias summary for included studies (Green: low risk of bias; yellow: unclear risk of bias; red: high risk of bias) [40–51].

### 3.3. Primary Outcomes

The results of the meta-analysis by pooling 12 studies using a random effects model showed that II intervention groups had a significant increase in PA compared to the control groups (SMD = 0.31, 95% CI: 0.12, 0.50, $p < 0.001$) (see Figure 4). The effect sizes for individual studies ranged from 0 to 1. The value of $I^2$ showed the presence of moderate heterogeneity. Figure 4 presents the results of the meta-analysis. From the funnel plot (see Figure S1), we can see the asymmetry, but according to the results of the Egger test ($p > 0.1$), showing that no significant small sample bias event occurred.

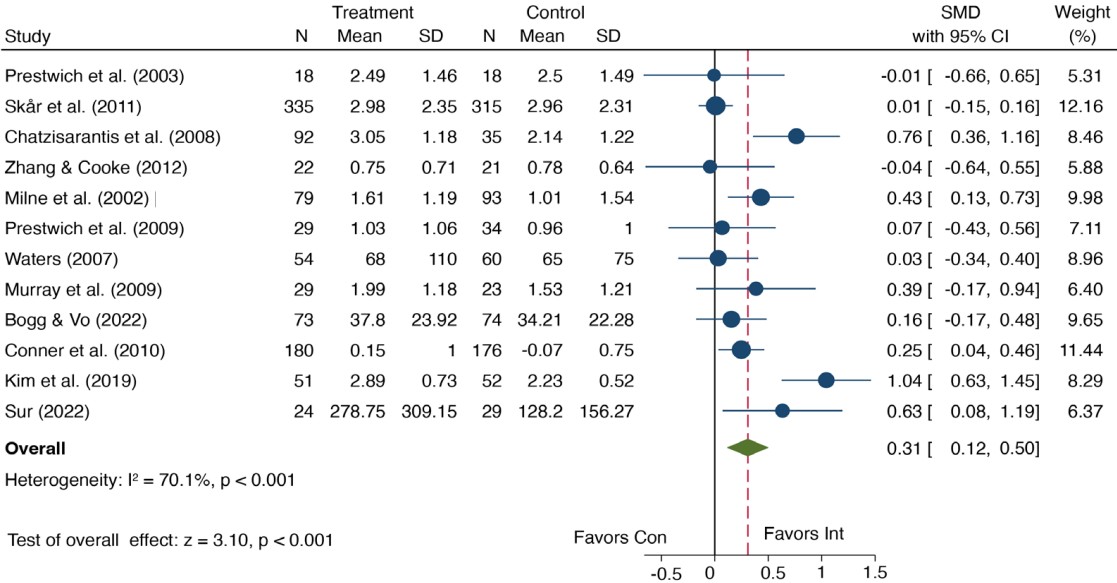

**Figure 4.** Meta-analysis of Effects of Implementation Intentions Interventions on PA among University Students Versus Control [40–51].

### 3.4. Subgroup Analysis

The results of the predetermined exploratory subgroup analyses based on the role of the five moderating variables are presented in Table 2. Interestingly in the subgroups of publication years, participants, and reinforcement, consistent findings were observed

that the intervention groups in each group of these subgroups significantly increased university students' PA compared to their respective control groups. However, the groups with publication years after 2013, inactive university students, and with reinforcement had larger effect sizes. In addition, in subgroup analyses of the intervention strategy, and duration, it was observed that only a single group in the subgroup with II interventions significantly enhanced university students' PA compared to the control group: the AP group, the group with reinforcement, and the group with an intervention period of less than six weeks, respectively.

**Table 2.** Subgroup Analysis.

| Moderators | Category | Studies | Heterogeneity Test | | SMD and 95%CI | $p$ |
|---|---|---|---|---|---|---|
| | | | $p$ | $I^2$ (%) | | |
| **Publish Year** | ≥2013 | 3 | 0.004 | 82.2 | 0.60 (0.02, 1.17) | 0.042 |
| | <2013 | 9 | 0.020 | 55.9 | 0.22 (0.04, 0.39) | 0.015 |
| | Overall | 12 | <0.001 | 70.1 | 0.31 (0.12, 0.50) | 0.001 |
| | Between | | 0.216 | | | |
| **Participant** | General | 8 | <0.001 | 73.7 | 0.27 (0.04, 0.50) | 0.023 |
| | Inactive | 4 | 0.053 | 61.0 | 0.40 (0.05, 0.74) | 0.025 |
| | Overall | 12 | <0.001 | 70.1 | 0.31 (0.12, 0.50) | 0.001 |
| | Between | | 0.550 | | | |
| **Intervention Strategy** | AP | 9 | 0.004 | 64.9 | 0.36 (0.14, 0.57) | 0.001 |
| | AP + CP | 3 | 0.099 | 56.7 | 0.15 (−0.22, 0.52) | 0.418 |
| | Overall | 12 | <0.001 | 70.1 | 0.31 (0.12, 0.50) | 0.001 |
| | Between | | 0.349 | | | |
| **Reinforcement** | Yes | 5 | 0.008 | 71.1 | 0.45 (0.08, 0.83) | 0.019 |
| | No | 7 | 0.008 | 65.5 | 0.22 (0.01, 0.42) | 0.036 |
| | Overall | 12 | <0.001 | 70.1 | 0.31 (0.12, 0.50) | 0.001 |
| | Between | | 0.285 | | | |
| **Duration** | ≥6 weeks | 4 | 0.550 | 0.0 | 0.05 (−0.07, 0.18) | 0.407 |
| | <6 weeks | 8 | 0.004 | 66.0 | 0.42 (0.17, 0.67) | 0.001 |
| | Overall | 12 | <0.001 | 70.1 | 0.31 (0.12, 0.50) | 0.001 |
| | Between | | 0.010 | | | |

Notes: Significance at $p < 0.05$.

*3.5. Sensitivity Analysis*

In assessing the reliability of the results of the meta-analysis, the method of examining the variations in effect sizes by excluding the literature one by one was adopted in the STATA16 software. The analysis showed that the variation in effect size was insignificant, indicating that the synthesized results were robust (see Figure S2).

**4. Discussion**

This study represents the first systematic review and meta-analysis to comprehensively evaluate the effects of IIs to promote PA among university students. In light of the inconsistent findings reported in the literature regarding the efficacy of II interventions on PA among university students, this study conducted a meta-analysis of 12 high-quality studies. The results revealed that II interventions positively impacted PA among university students, with small to medium effect sizes according to Cohen's criteria [52]. Exploratory subgroup analyses were conducted to determine the impact of various moderating variables. These analyses revealed significant intervention effects across all groups in the subgroups of publication year, participant characteristics, and reinforcement. However, significant effects were only observed in the AP group of the intervention strategy subgroup and the group with less than six weeks of follow of the duration subgroup.

The results of this study provide compelling evidence supporting the effectiveness of II interventions in promoting PA among university students. These findings are consistent

with previous studies conducted in the general population [30–32] and chronic disease groups [33], which have also reported significant positive effects of II interventions on PA. Another similarity with these studies is the observation of small to medium effect sizes for the significant effects.

In contrast to the current study's findings, the earlier meta-analysis conducted by Gollwitzer and Sheeran [24] reported a medium to large, pooled effect size after combining the effect sizes of 96 studies that investigated the application of II interventions for various health behaviors. A possible reason for this gap is the variation in the effects of II interventions across different health behavior changes. Another possible explanation is that PA is a complex behavior, and the role of self-regulatory variables from intention to behavior occurrence may be more susceptible to other covariates, such as goal multiplicity, emotional status, and exercise habituation [53]. The initiation of PA is an inherently intricate process, where individuals navigate positive outcome expectancies, such as improved functional status and weight loss, and negative outcome expectancies, such as fatigue, physical stress response, and time costs, during the process of forming intentions to implementing behavior [54,55]. While self-regulatory variables such as IIs can mediate and moderate this process, their effectiveness may be reduced when the strategies lack sufficient specificity and detail, such as neglecting to specify when, where, and how to implement the behavior, or fails to address obstacles encountered during the formation of the behavior [56]. Furthermore, the results of the present study are inconsistent with the findings of Silva et al. [57], whose study did not observe a significant improving effect of II interventions on PA. The more rigorous quality control of the literature and the lower heterogeneity reflected by $I^2$ in the present study may provide strong support for the reliability of the current findings.

Implementation intentions are a psychological process that involves planning the details of prospective behavior [13]. The applications of II interventions typically involve sessions of planning, reinforcement, and outcome measurement [26]. As II interventions are commonly implemented in a single session at the beginning, with subsequent repetitions serving as reinforcements, it is more appropriate to use the results of the last measurement point after the interventions as the values for combined effect sizes in this study. This pooling approach provides valuable insights for assessing II interventions' long-term and sustainable effectiveness for PA. The small to medium effect sizes combined in this review may provide strong evidence that II interventions can consistently improve PA in university students.

This study conducted a five-group subgroup analysis and yielded valuable findings to investigate the factors that influence heterogeneity and examine the variability of physical activity in different scenarios and contexts following II interventions. Unlike previous reviews that have addressed a broader range of participants, this study focused exclusively on university students. Subgroup analysis based on the recruitment criteria for participants in the included studies identified general and inactive groups. The results showed significant improvements in PA for both groups, with II interventions having a larger effect size in the inactive university student group. This finding suggests that II interventions may be particularly effective for inactive university students. Previous studies have identified factors such as lack of interest (insufficient motivation) [58], low self-efficacy [12], time management barriers [59], and lack of persistence [60] as the main reasons that affect university students' participation in PA. The intrinsic mechanism of IIs is a psychological simulation process that involves planning and responding to the details of prospective physical activity participation [61,62]. This pre-set psychological process can facilitate the simulation of initiating, implementing, and maintaining a perceived goal [63]. Self-efficacy has long been considered an essential psychological determinant of participation in PA [64,65], and its improvement is often highly correlated with past experience (including both direct and vicarious experiences) [66]. The pre-setting and responding to scenarios (If-then) in II intervention is an implementation of this experience [14]. Thus, the improvement of PA in inactive university students under II interventions may be closely related to the enhancement of self-efficacy.

Surprisingly, in the subgroup analysis of interventions, only the AP group showed a significant intervention effect, while the combined interventions of AP and CP did not show a significant impact. One possible reason for this phenomenon is that II intervention, as one of BCTs, may be more effective when the content is focused on a single point to fully activate the behavioral "switch" [67]. Too much intervention content may cause confusion and distraction. Many studies included in this review mentioned the original "If-then" concept of IIs, which is most consistent with the core elements of CP based on conditional triggering. However, none of the studies included in this review designed interventions around this concept, which may also suggest that the specific planning strategy of AP may be more effective. Nevertheless, combined intervention with multiple measures is very promising for behavior change [68]. During the implementation of interventions, how CP measures can be combined with AP to achieve the effectiveness of the combined intervention is a question worth exploring.

As expected, the subgroup analysis of intervention duration identified larger effect sizes for shorter follow-up periods and insignificant effects for follow-up periods longer than six weeks. This finding suggests that the effects of II interventions on PA in university students' relapse over time. Relapse is a potential occurrence in health behavior change over time [69], and continuous reinforcement is vital to maintaining intervention effects [70]. This principle was confirmed in the subgroup analysis of groups with and without reinforcement measures, where the effect size of intervention effects was larger in groups with reinforcement measures after the intervention. Unfortunately, although some studies have observed better intervention effects under high-frequency reinforcement measures [51], the relationship between the frequency and mode of intervention reinforcement and intervention effects cannot be determined due to the limited number of included studies.

An interesting observation was made in the subgroup analysis of publication year: studies published after 2013 had larger effect sizes, reaching medium to high levels. This finding suggests that the effectiveness of II interventions for PA among university students is improving over time, indicating that this approach is maturing and has broad application prospects on university campuses.

Recent studies have achieved positive intervention effects by integrating II interventions with other BCTs [71,72], such as MCII [73]. Undoubtedly, as a necessary "bridge" between intention and behavior [13], IIs have broader application value in the field of PA. The application of e-health for promoting PA has emerged as a recent research hotspot in the field of health promotion [74,75]. However, there is limited research focused on improving PA through the integration of II interventions and ICTs. Although face-to-face pen-based records are commonly employed to carry out IIs, delivery modes such as mobile phone text messaging, social media, email, and websites in the field of e-health might offer enhanced convenience and efficiency of II intervention without incurring additional costs. Furthermore, university campuses are recognized as highly technologically advanced environments, and university students are inherently digital natives [76]. Hence, combining IIs with digital technology to promote PA and other healthy behaviors holds significant and sustainable prospects within university campuses and is a field worth exploring for health practitioners and education professionals [77].

Although this study included high-quality RCT literature in determining the effectiveness of IIs in promoting PA among university students for the first time, several limitations need to be addressed in future research. First, although the number of included studies exceeded the threshold for meta-regression, the number of studies is still relatively small. To avoid statistical bias, only exploratory subgroup analysis was performed when investigating the role of moderators. Second, all included studies used self-reported questionnaires or items to measure PA, which may result in varying results. It is hoped that more objective measurement tools, such as pedometers and accelerometers, will be applied in future PA research. Third, the delivery modes of II interventions may significantly impact the results. However, as only two studies used online self-management interventions and the other ten

used interviewer-assisted interventions in sessions, we did not conduct a subgroup analysis in this regard. It is anticipated that more high-quality studies will be included in future meta-analyses. Fourth, this study focused on a single population of university students to reduce heterogeneity due to participants. However, combining multiple types of PA outcomes poses a challenge to accurately interpreting the results. It is expected that more trials will focus on this field and adopt a standardized approach to measure PA, ensuring the stability and reliability of the evidence.

## 5. Conclusions

This study confirmed that II interventions could significantly promote PA among university students, particularly for inactive university students, when using a single AP intervention, with reinforcement and when the follow-up period is less than six weeks. These findings provide theoretical support and targeted intervention guidance for improving PA among university students. II interventions are cost-effective and easy to implement within university campuses, contributing to the generation of more sustainable effects. Future research should focus on combining IIs with technology, integrating the psychological determinants of behavior change inherent in IIs with ICT's media and presentation methods to achieve more refined planning interventions. In addition, personalized and multiple intervention models are also worth exploring and implementing in future research.

**Supplementary Materials:** The following materials are available online at https://www.mdpi.com/article/10.3390/su151612457/s1, search strategy; Figure S1: funnel plot; Figure S2: sensitivity analysis; and PRISM checklist.

**Author Contributions:** Conceptualization, S.P. and A.T.O.; methodology, S.P., J.L. and Z.Z.; formal analysis, S.P. and A.Z.K.; investigation, S.P. and F.Y.; writing—original draft preparation, S.P.; writing—review and editing, S.P. and X.Z.; supervision, A.T.O. and A.Z.K.; project administration, S.P.; funding acquisition, S.P. All authors have read and agreed to the published version of the manuscript.

**Funding:** This research is supported by funding from the Discipline Building Reserve Project of Hohai University (2023). This research was also supported by the Fundamental Research Funds for the Central Universities (Hohai University, B230207048).

**Institutional Review Board Statement:** Not applicable.

**Informed Consent Statement:** Not applicable.

**Data Availability Statement:** Data generated or analyzed during this study are included in this published article or in the data repositories listed in the references.

**Acknowledgments:** We would like to thank the School of Educational Studies of Universiti Sains Malaysia (USM) for facilitating this study. We would also like to thank all the authors of the literature included in this study.

**Conflicts of Interest:** The authors declare no conflict of interest.

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
