# Peer review of "Meta-Analysis of Implementation Intentions Interventions in Promoting Physical Activity among University Students"

_sustainability, doi:10.3390/su151612457_

Round 1

Reviewer 1 Report

The study has a proper method for a systematic review and meta-analysis. The authors discussed all of their findings reported in the results section. I have no other suggestion. The paper is ready for publication in my view.

Author Response

Dear reviewer,

Thank you for your letter and for the reviewers' comments concerning our manuscript entitled " Evaluating the Effectiveness of Implementation Intentions Interventions in Promoting Physical Activity Among University Students: A Systematic Review and Meta-analysis " (sustainability-2454691). Those comments are all valuable and helpful for revising and improving our paper, as well as the essential guiding significance to our research. Thank you again for your high approval of our research work, and we will further optimize the manuscript according to other reviewers’ suggestions to meet publication requirements.

We are grateful for the reviewer's time and effort in evaluating our work, and we are committed to addressing their concerns and improving the manuscript accordingly. We welcome any further suggestions or guidance that will help us enhance the quality and impact of our research.

Thank you for your consideration.

Best wishes for you

Sincerely, yours

Reviewer 2 Report

Dear authors,

My biggest concern is that, your manuscript is not based on a solid research/knowledge/theoretical gap.

In my opinion, nothing novel is offered as the manuscript as it is not designed to seal any significant  gap.

I am really concern about the the possibility for your manuscript to receive citations.

The analysis was conducted in detail, which is a positive aspect.  Yet, I can't see any significant of this work.

Thank you.

Author Response

Dear reviewer,

Thank you for your letter and for the reviewers' comments concerning our manuscript entitled " Evaluating the Effectiveness of Implementation Intentions Interventions in Promoting Physical Activity Among University Students: A Systematic Review and Meta-analysis " (sustainability-2454691). Those comments are all valuable and helpful for revising and improving our paper, as well as the essential guiding significance to our research. We have studied the comments carefully and have made corrections which we hope to meet with approval. Regarding the concerns raised by the reviewer, we have made every effort to provide thorough explanations to address them. Revised portions are marked in highlighted red (normal revision) in the paper using the track change mode of MS Word. The main corrections in the paper and the response to the reviewer's comments are in the attachment.

Once again, we express our heartfelt gratitude to the reviewer for their selfless dedication and diligent work.

Best wishes,

Sincerely, yours

Reviewer 3 Report

- The abstract should be more concise. At the moment it is quite redundant and not clear. Too many acronymous confuse the reader.

- please avoid the use od acronyms in the paper when there are not necessary

- please specify from which database you have downloaded the papers (webesco, scopus?)

- the analysis is well developed, but a more accurate discussion is needed.  For example provide a section on possible future research questions or gaps in the literature!

the quality of the English is fine. The flow is a bit scuttled and can be improve by a polishing on the style of writing 

Author Response

Dear reviewer,

Thank you for your letter and for the reviewers' comments concerning our manuscript entitled " Evaluating the Effectiveness of Implementation Intentions Interventions in Promoting Physical Activity Among University Students: A Systematic Review and Meta-analysis " (sustainability-2454691). Those comments are all valuable and helpful for revising and improving our paper, as well as the essential guiding significance to our research. We have studied the comments carefully and have made corrections which we hope to meet with approval. Regarding the concerns raised by the reviewer, we have made every effort to provide thorough explanations to address them. Revised portions are marked in highlighted red (normal revision) in the paper using the track change mode of MS Word. The main corrections in the manuscript and the response to the reviewer's comments are in the attachment.

We are grateful for the reviewer's time and effort in evaluating our work, and we are committed to addressing their concerns and improving the manuscript accordingly. We welcome any further suggestions or guidance that will help us enhance the quality and impact of our research.

Thank you for your consideration.

Best wishes for you

Sincerely, yours

Reviewer 4 Report

Please immediately revise your research manuscript according to the suggestions for improvement that I wrote in your manuscript!

Please immediately revise your research manuscript according to the suggestions for improvement that I wrote in your manuscript!

Author Response

Dear reviewer,

Thank you for your comments concerning our manuscript entitled "Meta-analysis of Implementation Intentions Interventions in Promoting Physical Activity among University Students"(Revised Title) (Manuscript ID: sustainability-2454691). Those comments are all valuable and helpful for revising and improving our paper, as well as the essential guiding significance to our research. We have studied the comments carefully and have made corrections which we hope to meet with approval. Revised portions are marked in highlighted red (normal revision) in the paper. The main corrections in the paper and the response to the reviewer's comments are appended to the attachment.

Best wishes

Sincerely, yours

Peng Sanying
